# Apoptosis mapping in space and time of 3D tumor ecosystems reveals transmissibility of cytotoxic cancer death

Irina Veith [1,2‡], Arianna Mencattini [3‡], Valentin Picant [2], Marco Serra [4], Marine Leclerc [5], Maria Colomba Comes [3], Fathia Mami-Chouaib [5], Jacques Camonis [2], Stéphanie Descroix [4], Hamasseh Shirvani [1], Fatima Mechta-Grigoriou [2], Gérard Zalcman [2,6], Maria Carla Parrini [2‡*], Eugenio Martinelli [3‡*]

**1** Institut Roche, 4 cours de l'Ile Seguin, Boulogne-Billancourt, France, **2** Institut Curie, INSERM U830, Stress and Cancer Laboratory, PSL Research University, Paris, France, **3** Department of Electronic Engineering, University of Rome Tor Vergata, Rome, Italy, **4** Institut Curie, CNRS UMR168, Laboratoire Physico Chimie Curie, Institut Pierre-Gilles de Gennes, PSL Research University, Paris, France, **5** INSERM UMR 1186, Integrative Tumor Immunology and Immunotherapy, Gustave Roussy, Fac. de Médecine—Univ. Paris-Sud, Université Paris-Saclay, Villejuif, France, **6** CIC INSERM 1425, Thoracic Oncology Department, University Hospital Bichat-Claude Bernard, Université de Paris, Paris, France

‡ IV and AM share equal first author contribution. MCP and EM share equal last author contribution
* martinelli@ing.uniroma2.it (EM); maria-carla.parrini@curie.fr (MCP)

**Data Availability Statement:** All relevant data are within the manuscript and its Supporting Information files.

## Abstract

The emerging tumor-on-chip (ToC) approaches allow to address biomedical questions out of reach with classical cell culture techniques: in biomimetic 3D hydrogels they partially reconstitute *ex vivo* the complexity of the tumor microenvironment and the cellular dynamics involving multiple cell types (cancer cells, immune cells, fibroblasts, *etc.*). However, a clear bottleneck is the extraction and interpretation of the rich biological information contained, sometime hidden, in the cell co-culture videos. In this work, we develop and apply novel video analysis algorithms to automatically measure the cytotoxic effects on human cancer cells (lung and breast) induced either by doxorubicin chemotherapy drug or by autologous tumor-infiltrating cytotoxic T lymphocytes (CTL). A live fluorescent dye (red) is used to selectively pre-stain the cancer cells before co-cultures and a live fluorescent reporter for caspase activity (green) is used to monitor apoptotic cell death. The here described open-source computational method, named STAMP (spatiotemporal apoptosis mapper), extracts the temporal kinetics and the spatial maps of cancer death, by localizing and tracking cancer cells in the red channel, and by counting the red to green transition signals, over 2–3 days. The robustness and versatility of the method is demonstrated by its application to different cell models and co-culture combinations. Noteworthy, this approach reveals the strong contribution of primary cancer-associated fibroblasts (CAFs) to breast cancer chemo-resistance, proving to be a powerful strategy to investigate intercellular cross-talks and drug resistance mechanisms. Moreover, we defined a new parameter, the 'potential of death induction', which is computed in time and in space to quantify the impact of dying cells on neighbor cells. We found that, contrary to natural death, cancer death induced by chemotherapy or by CTL is transmissible, in that

**Funding:** This work was supported by Fondation ARC pour la Recherche sur le Cancer, PGA1 RF20180206991 to MCP (https://www.fondation-arc.org/). IV is supported by a CIFRE fellowship founded in part by National Association for Research and Technology (ANRT) (http://www.anrt.asso.fr/fr), on the behalf of the French Ministry of Higher Education and Research, and in part by Institut Roche (https://www.roche.fr/fr/innovation-recherche-medicale/institut-roche.html). Institut Roche participated to the thesis supervision of IV. The funders had no other role in study design, data collection and analysis, decision to publish, or preparation of the manuscript.

**Competing interests:** IV and HS are Roche employees. IV, AM, MCP, EM filed a patent on the STAMP method.

it promotes the death of nearby cancer cells, suggesting the release of diffusible factors which amplify the initial cytotoxic stimulus.

## Author summary

The tumor microenvironment (TME) is a very complex cellular ecosystem, composed of the cancer cells (carrying the disease-causing genetic alterations), immune cells and other stromal cells (such as fibroblasts), which contribute to disease progression and drug responses. Here, we investigated these complex cellular dynamics by reconstituting the tumor ecosystems in a very controlled manner within microfluidic devices, with multiple cell populations, generating the so-called 'tumors-on-chip', which can be visualized by video-microscopy and treated with anti-cancer drugs. The resulting videos contain a huge amount of information that requires advanced computational approaches to be extracted. In this work, we developed a novel method, named STAMP, that precisely measures the kinetics and the spatial maps of cancer cell deaths within tumor-on-chip. Two case studies are presented: breast cancer cells upon chemotherapy treatment (doxorubicin) and lung cancer cells upon killing by specific immune cells (tumor-infiltrating cytotoxic T lympho-cytes). We generated spatio-temporal maps on cancer death uncovering unsuspected rela-tions between death events. This indicates that dying cancer cells might release soluble factors that induce death of neighbor cancer cells. The STAMP method was suitable to study the capacity of fibroblasts to promote resistance of cancer cells to chemotherapy.

## Introduction

Recent advances in microfluidics and microfabrication inspired new solutions to reproduce *ex vivo* 3D microarchitectures on chip imitating characteristics of organ functional units and of tumor microenvironments (TME). This established the basis for the technology of organ-on-chip (OoC) [1–3] and tumor-on-chip (ToC) [4–6]. The OoC/ToC technology offers numerous advantages, such as tight control of biological and physicochemical conditions (cell types, 3D biomimetic hydrogel, biochemical environment), real-time observation of cellular dynamics, miniaturization (few cells and little reagent are needed), fast results, and lower costs. Despite this huge potential, so far, the ToC use has been restrained to specialized laboratories and has not reached the broad community of cancer researchers. Several promising applications in basic and translational research, as well as in clinics, have been proposed, but their implemen-tation clearly requires further developments. In particular, a major bottleneck of ToC technol-ogy is the lack of standardized user-friendly computer tools to process, analyze, and fully exploit the rich information generated by ToC imaging. The integration of advanced image analysis tools and deep learning methods is expected to foster new powerful solutions to this problem.

Our previous works demonstrated the feasibility of reconstituting on-chip various tumor ecosystems, composed of up to four cell types (cancer cells, immune cells, cancer-associated fibroblasts [CAF], and endothelial cells), which can be treated with various anti-cancer drugs, including standard chemotherapies and targeted therapies (*e.g.*, trastuzumab) [7–9]. The vid-eos faithfully monitor the cancer death, in time and space, upon these various treatments. In this work we develop, validate and apply a novel computational method, named STAMP, to

automatically extract the temporal kinetics and the spatial maps of cancer death in ToC cultures.

Thanks to their capacity to capture the cell death kinetics, image analysis approaches are progressively replacing the historical end-point cytotoxic assays, such as the luminescent detection of ATP [10] or the $^{51}$Cr-release assay [11]. For example, a recent work combines live/dead cell markers and mathematical modeling to achieve a high-throughput analysis of cell death kinetics with over 1800 bioactive compounds [12]. Similarly, image analysis algorithms to measure cytotoxic or apoptotic index are commercially available (*e.g.*, IncuCyte-Essen BioScience or NanoLive). A real-time bio-imaging cytotoxic assay has been proposed for 96-well microplate [13]. All these software tools have been conceived to work in 2D settings, with focus on temporal information. Recently, an ingenious 96-well microfluidic platform was developed to perform bio-imaging cytotoxic assay in 3D gels [14]. Since 3D microfluidic devices allow to keep confined the cells as well as their released soluble factors, they are appropriate to investigate the consequences of each death event on surrounding cells. For this purpose, we focused on analysis strategies to extract not only the temporal information, but also the spatial information of cancer death events. STAMP introduces the new concept of 'potential of death induction', by calculating the induction that each death region (defined as 'object') produces on the surrounding regions, with respect to their mutual distances and to their temporal relationships. The combination of measures both in time and in space allowed us to conduct an original apoptosis analysis that accounts not only for the number of death events and their kinetics, but most shrewdly for their spatial distribution in the 3D confined environments of ToC cultures.

## Results

### Imaging strategy to monitor cancer death in tumor-on-chip (ToC) co-cultures

In order to generate 3D tumor-on-chip (ToC) co-cultures, we used commercially available microfluidic devices in plastic (AIM-Biotech), that were imaged under an inverted video-microscope with controlled $CO_2$ (5%) and temperature (37˚C) for 2–3 days. Cells were embedded in a 3D biomimetic collagen gel and injected in the 3.41 mm$^3$ chamber of the microfluidic device.

For this work we generated mono-cultures (cancer cells only) and two kinds of bi-cultures (cancer cells with immune cells, cancer cells with CAFs). For all experiments, a live fluorescent dye (CellTrace, red) was used to selectively pre-stain the cancer cells before cultures on-chip, and a live fluorescent reporter for caspase activity (CellEvent Caspase-3/7, green) was added to on-chip culture medium to monitor apoptotic death. No matter the degree of co-culture complexity, the cancer death detection was achieved by monitoring the red to green signal transitions.

### Description of the STAMP method

A computational strategy was developed to automatically and objectively monitor, in time and in space, the events of apoptotic cancer cell deaths, *i.e.* the red to green signal transitions (Fig 1, see Materials and Methods for details). The software was named STAMP, from S̲patiotemporal a̲poptosis m̲apper.

Briefly, from multi-channel videos of ToC co-cultures, the cancer cells are localized and tracked in the red channel. The dying cancer cells are identified in the green channel, after signal normalization and thresholding. The death signal is modeled to vanish during T time frames. Then, the spatiotemporal features of all death signals are integrated in a unique map, from which a novel parameter, named potential of death induction ($P_{death}$), is computed over time.

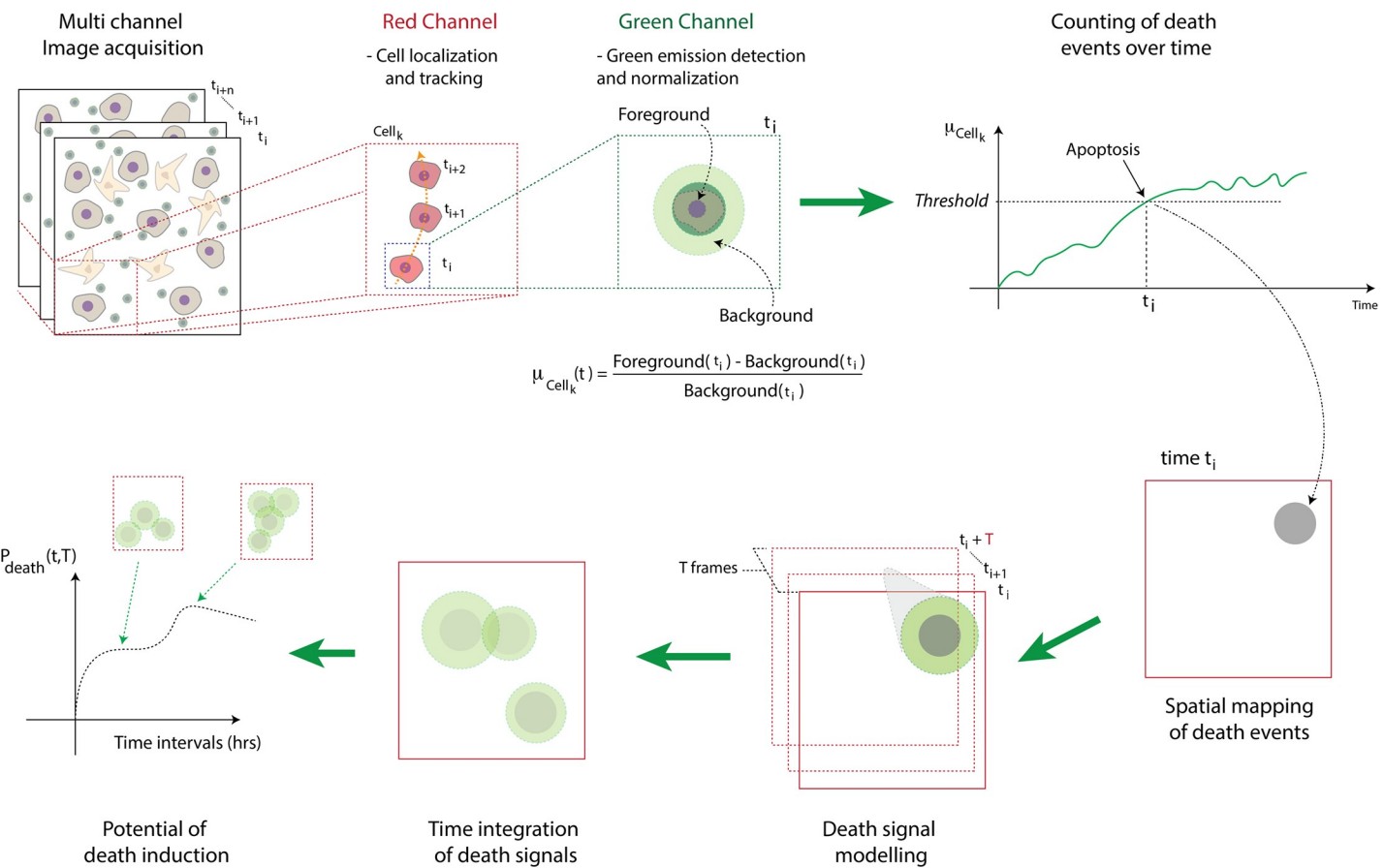

$$\mu_{Cell_k}(t) = \frac{Foreground(t_i) - Background(t_i)}{Background(t_i)}$$

**Fig 1. Description of the STAMP method.** From multi-channel videos of ToC co-cultures, the cancer cells (pre-stained in red) are localized and tracked in the red channel. The dying cancer cells (becoming green because of the caspase reporter) are identified in the green channel, after signal normalization and thresholding. The method monitors cancer cell deaths both in time and in space. The death signal is modeled to vanish during T time frames. Then, the spatiotemporal features of all death signals are integrated in a unique map, from which a parameter, named potential of death induction ($P_{death}$), is computed over time (see Materials and Methods for details).

Several output measurements were extracted and used for the following analysis. First, the apoptotic rate, *i.e.* the percentage of cancer cells dying within a certain $T_{LAG}$ time interval (4 to 10 h, in this study), calculated using the number of cells at the beginning of each time interval as starting reference. Second, the overall survival, *i.e.* the percentage of cancer cells alive over time, which is calculated using the number of cells at the beginning of the experiment as starting reference and therefore takes into account both cell death and proliferation. Third, the spatiotemporal map of death events, integrating the information of when and where all deaths occur. Fourth, the potential of death induction ($P_{death}$) within a $\tilde{T}$ time over the entire field of view and experimental time, measuring the capacity of dying cells to promote the death of nearby living cells in the 3D experimental setting.

## Application to quantify chemotherapy-mediated cytotoxicity in breast-cancer-on-chip cultures

First, we applied the STAMP method to analyze the response of a standard cell model, the triple-negative breast cancer MDA-MB-231 cells, to a standard chemotherapy drug, doxorubicin (Fig 2 and S1 and S2 Movies). To achieve a moderate killing, we chose a doxorubicin

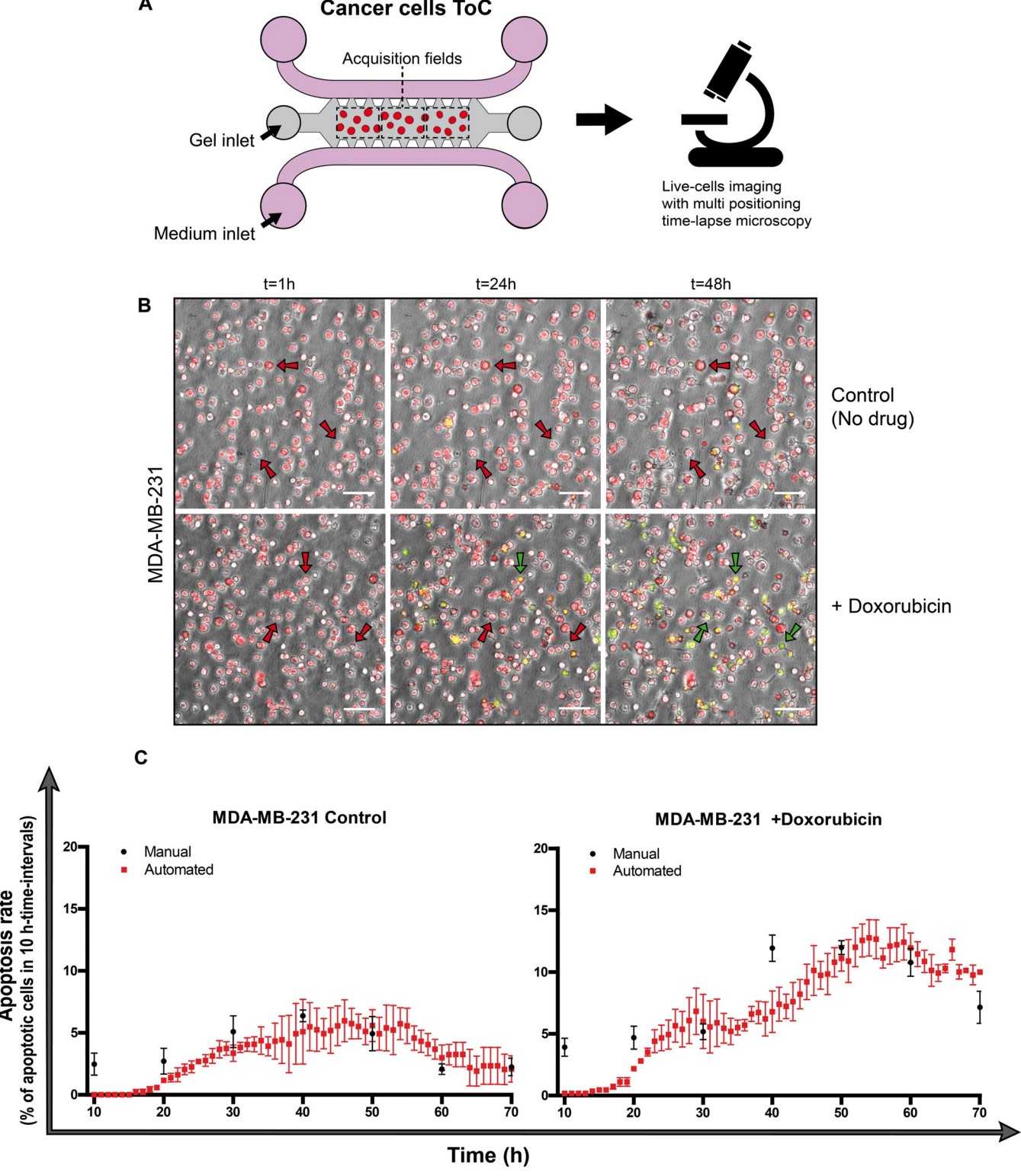

**Fig 2. Chemotherapy-mediated cytotoxicity in breast-cancer-on-chip cultures. A.** Experimental design: breast cancer MDA-MB-231 cells are embedded in a collagen matrix in the central chamber of the chip; cells are live-imaged in transmission channel and fluorescence channels (red and green) every hour for 72 h, without or with doxorubicin (1 μM). **B.** Representative images of MDA-MB-231 cells after 1 h, 24 h and 48 h of culture on-chip, without drug (uppers panels) or with doxorubicin (lowers panels). Red arrows indicate living cells, whereas green arrows point at apoptotic cells. Scale bar, 100 μm. **C.** Time-course quantifications of the apoptosis rate, calculated in 10 h-time-intervals, showing the comparison between manual counts (black rounds) and automatic counts

(red squares, $T_{LAG}$ = 10 h), without (left) or with (right) doxorubicin. The means +/- SEM of 3 measurements on 3 view fields from the same experiment were calculated automatically with STAMP every hour and manually every 10 hours. For statistical analysis, the measurements for each of the two conditions (manual and automated) were assembled regardless of the time variable. The non-parametric Mann-Whitney test was applied, since the data did not pass the Shapiro-Wilk normality test, and the difference resulted not significant.

concentration of 1 μM, which is slightly lower than the IC50 of doxorubicin for MDA-MB-231 cells (2.8 μM), that we previously measured in standard 2D dishes.

In order to benchmark the accuracy of the automated method, we compared the values obtained by the algorithm with those obtained by manual counting. Manual and automated counting of the apoptosis rate reached numbers in very similar ranges (Fig 2C). Even if the numbers were not identical for all time points, the curve trends were very similar. We observed that the algorithm had a tendency to underestimate the apoptotic cells at early time-points. This might be due to the fact that the green channel threshold, that is used to detect apoptotic events (Fig 1), is automatically determined over the 72 h. Early, weak signals may be lost because under the threshold. Conversely, a human operator cannot discriminate between weak and strong green emissions and therefore may assign apoptotic events to any green emission increase. Importantly, we could reach a much higher (10-fold) time resolution for automated measurements (every hour) than for manual measurements (every 10 hours). Therefore, STAMP is a *bona fide* automated method to measure apoptosis rate.

In control MDA-MB-231 cells without drug, the basal apoptosis rate in 10 h-time-intervals ($T_{LAG}$ = 10 h) fluctuated around 5% during the experiment time (72 h), meaning that roughly 5% of the cells died every 10 h (Fig 2C, left). In doxorubicin-treated cells, the death rate remained at basal level during the first 20 h of treatment, then after 20 h of doxorubicin exposure the death rate increased up to more than 10% (Fig 2C, right). Therefore, the time-resolved STAMP analysis revealed that the speed of cytotoxic response to doxorubicin increases with the time in this 3D on-chip setting.

## Application to quantify T-cell mediated cytotoxicity in lung-cancer-on-chip cultures

Next, we challenged the STAMP method with a more complex situation in which a non-small cell lung cancer (NSCLC) cell line (IGR-Pub) was co-cultured with an autologous CTL clone (P62) that was isolated from tumor-infiltrating lymphocytes (TIL) and selected to recognize and kill the cognate target [15] (Fig 3 and S3 and S4 Movies). To achieve a moderate killing, we chose a 1:1 effector (CTL) to target (cancer) cell (E:T) ratio.

The algorithm could accurately distinguish the prestained cancer cells from the unstained T cells, and again all the values obtained by the algorithm were very similar to those obtained by manual counting (Fig 3C).

The basal apoptosis rate of IGR-Pub cells in 10 h-time-intervals ($T_{LAG}$ = 10 h) was very low (around 2%) during the experiment time (48 h). The presence of the T cells immediately induced a significant death rate (around 10%); after 30 h of co-cultures the apoptosis rate dramatically increased (up to 30%). Interestingly, similarly to what observed for cytotoxic response to doxorubicin (Fig 2C, right), the speed of cytotoxic response to CTL appears to increase with the time as well (Fig 3C, right).

We further characterized, in a separate experiment, the real-time dependency of T-cell mediated cytotoxicity on T-cell density by using different E:T ratios with a better time resolution ($T_{LAG}$ = 4 h) (Fig 4). At low T-cell density, 1:10 E:T ratio, the apoptosis rate was not different from the control without T cells. A mild killing started to appear at 1:2 E:T ratio, but only at 1:1 ratio, an efficient killing could be detected (Fig 4A). Interestingly, there was not a linear

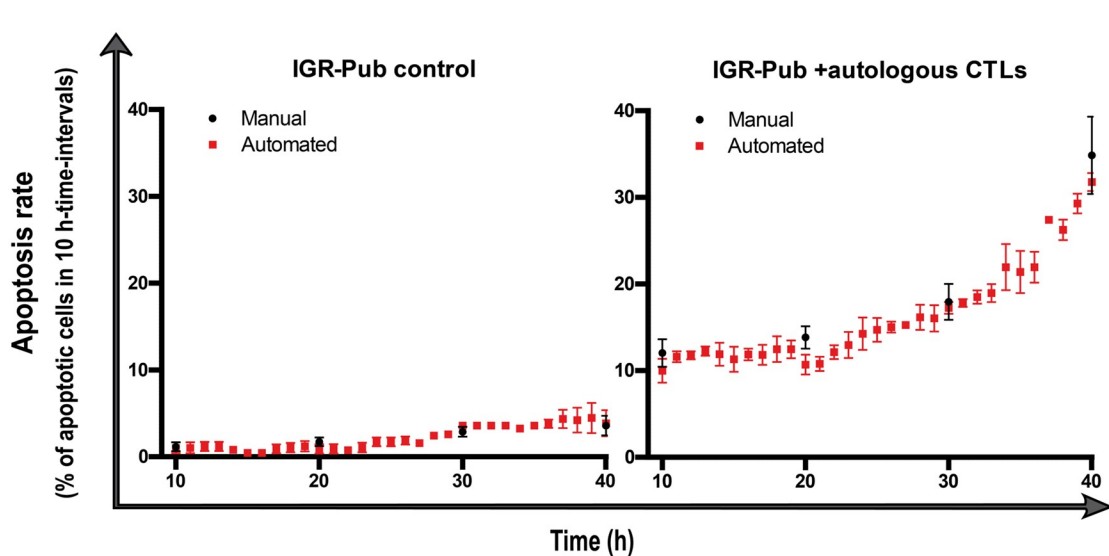

**Fig 3. T-cell mediated cytotoxicity in lung-cancer-on-chip cultures. A.** Experimental design: the lung cancer IGR-Pub cells are embedded in a collagen matrix in the central chamber of the chip, alone or together with autologous CTLs (P62 clone) at 1:1 effector to target cell (E:T) ratio; cells are live-imaged in transmission channel and fluorescence channels (red and green) every hour for 48 h. **B.** Representative images of IGR-Pub cells after 1 h, 24 h and 48 h of culture on-chip, alone (uppers panels) or with autologous T cells (lowers panels). Red arrows indicate living cells, whereas green arrows point at apoptotic cells. Blue arrows point at CTLs. Scale bar, 100 μm. **C.** Time-course quantifications of the apoptosis rate, calculated in 10 h-time-intervals, showing the comparison between manual counts (black rounds) and automatic counts (red squares, $T_{LAG}$ = 10 h), without (left) or with (right) T cells. The means +/- SEM of 3 measurements on 3 view fields from the same experiment were calculated automatically with STAMP every hour and manually every 10 hours. For statistical analysis, the measurements for each of the two conditions (manual and automated) were assembled regardless of the time variable. The non-parametric Mann-Whitney test was applied, since the data did not pass the Shapiro-Wilk normality test, and the difference resulted not significant.

proportionality between density and killing capacity of T cells, suggesting threshold effects. Again, the death rate increased with the time of co-cultures, despite the fact that after 2 days on chip the T cell viability was reduced; as assessed from the videos by manual counting, during the 2-day experiment time 19.7 ± 3.1% of T cells showed green fluorescence, indication of apoptotic death.

Consistently, the overall survival curves of cancer cells, which takes into account the balance between cell death and cell proliferation (the on-chip IGR-Pub doubling time being approximately 5 days), showed a detectable T-cell mediated killing only for the 1:2 and 1:1 ratios (Fig 4B), with a near 80% and 40% overall survival respectively, after 48-h co-cultures.

## Cancer-associated fibroblasts promote chemo-resistance in breast-cancer-on-chip

Having established that STAMP is accurate to monitor cancer death within co-cultures of cancer and T cells, we moved to bi-cultures of cancer cells and CAFs. CAFs are a major component of the stroma which is crucial for tumor progression; in NSCLC tumor-stroma ratio could be used as prognostic factor for survival [16]. Since it is well established that CAFs contribute to chemo-resistance in various cancer types [17–22], we assessed the capacity of ToC to recapitulate *ex vivo* the CAF impact on doxorubicin resistance by co-culturing primary breast CAFs [8,23] with the breast cancer MDA-MB-231 cells (S5 and S6 Movies). The addition of CAFs (1,6 CAF:cancer cell ratio) slightly increased the basal MDA-MB-231 apoptosis rate. Interestingly, when CAFs were added, the doxorubicin-dependent apoptosis of MDA-MB-231 cells was completely impaired, indicating a protective role of CAF against chemotherapy (Fig 5). These results indicate that ToC technology and STAMP quantifications will be very valuable to study the mechanisms underlying stroma contribution to cancer progression and to drug resistance.

## Spatiotemporal analysis of cytotoxicity death reveals the release of pro-apoptotic signals

In addition to the temporal kinetics, the STAMP method allows to extract the localization of dying cells, to build cumulative spatial maps of time-integrated death events, and to compute a potential of death induction ($P_{death}$) that quantifies the capability of dying cells to promote the death of nearby cells (see Materials and Methods for mathematical details and S7 Movie for a representative STAMP output video). Of note, since the window size for the calculation of the $P_{death}$ is 283 μm x 283 μm, a 'nearby' cell would be a cell within this window.

The $P_{death}$ (see Eq (11)) combines in a unique parameter both spatial and temporal death induction effects. On one hand, the spatial distribution of regions with death events (dense or sparse) contributes to the final value of $P_{death}$ thanks to the dependency on the inverse of the mutual distances. On the other hand, the average value of the cumulative map MC, that takes

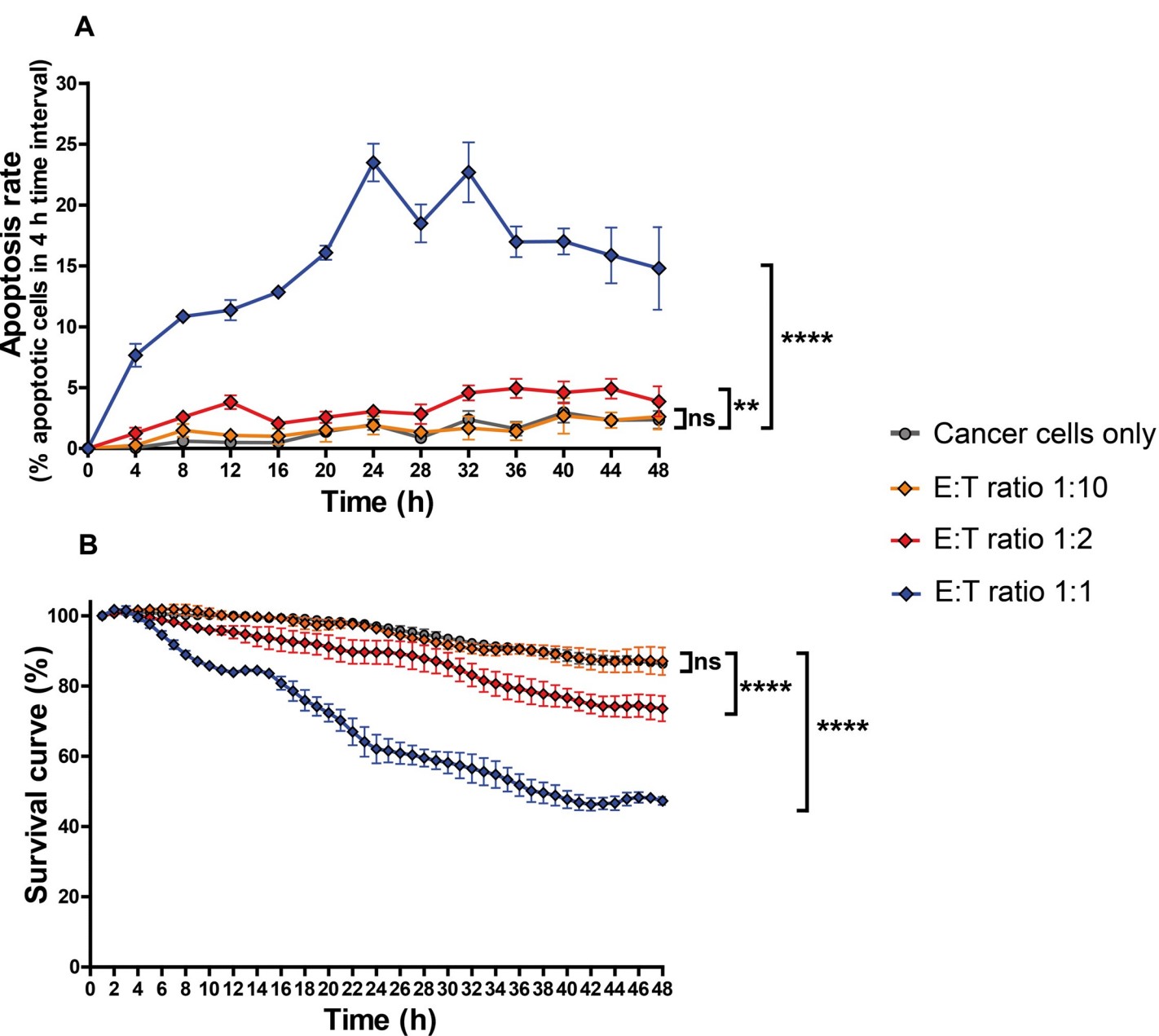

**Fig 4. Real-time dependency of T-cell mediated cytotoxicity on T cell density.** IGR-Pub lung cancer cells were co-cultured on-chip with autologous T cells at indicated E:T ratios. **A.** Time-course quantifications of the apoptosis rate, calculated in 4 h-time-intervals ($T_{LAG}$ = 4 h), for a total duration of 48 h. **B.** Survival curves of cancer cells within the ToC at 1-hr time resolution. For each $t_i$ the % of surviving cells, calculated with respect to the initial number of living cells, is the average of 3 hours, centered on $t_i$, to smoothen the fluctuations. The graphs show the means of 3 measurements on 3 view fields from the same experiment (+/- SEM). For statistical analysis, the measurements for each of the 4 conditions were assembled regardless of the time variable (n = 36 per condition). The different E:T ratio conditions were compared to the control condition of cancer cells only. In panel A, the data passed the Shapiro-Wilk normality test, therefore the unpaired t test was applied. In panel B, the data did not pass the Shapiro-Wilk normality test, therefore the non-parametric Mann-Whitney test was applied. **** indicates p<0.0001, ** indicates p<0.001, * indicates p<0.05 and ns indicates not significant.

into account the effect of the death wake in the temporal window $\tilde{T}$, contributes to $P_{death}$ thanks to the direct dependence on MC calculated for all paired death regions. In S1 Fig, we provide a simulated example for the $P_{death}$ calculation, to qualitatively explain how this parameter integrates both spatial and temporal properties.

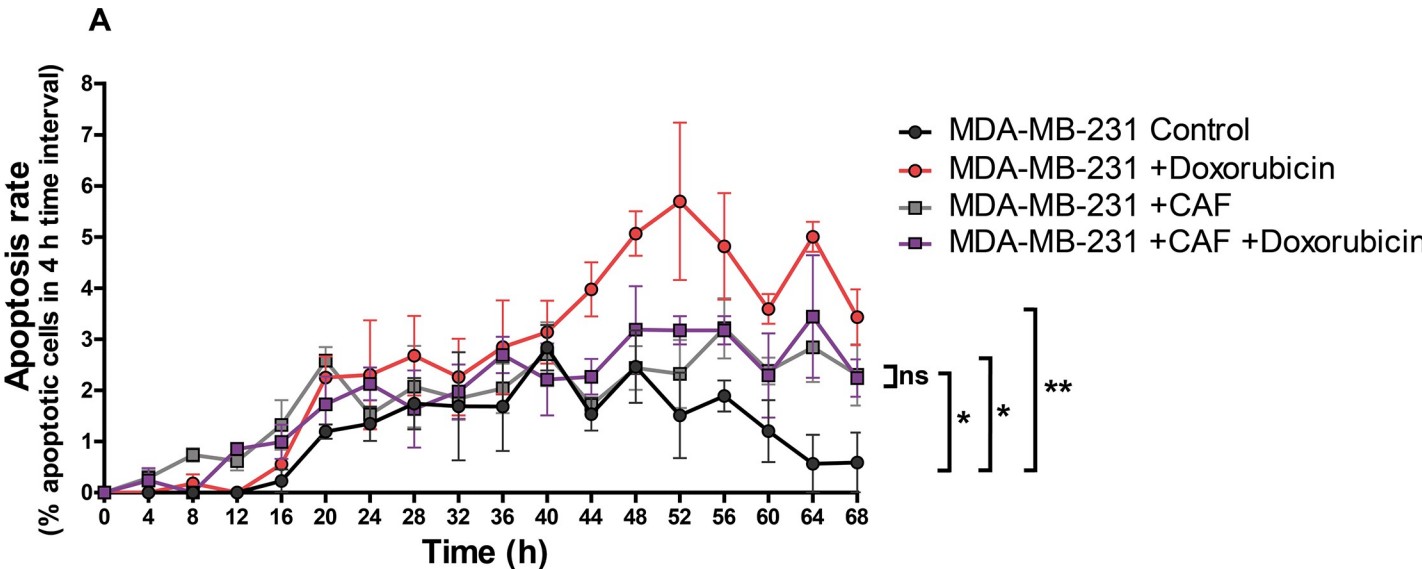

**Fig 5. Cancer-associated fibroblasts promote chemo-resistance in breast-cancer-on-chip.** MDA-MB-231 cancer cells were co-cultured on-chip +/- cancer-associated fibroblasts (CAFs) and +/- doxorubicin (1μM) treatment. Final CAF:cancer ratio was 1:6. The graph shows the time-course quantifications of the apoptosis rate, calculated in 4 h-time-intervals ($T_{LAG}$ = 4 h), for a total duration of 68 h. The graphs show the means of 3 measurements on 3 view fields from the same experiment (+/- SEM). The data illustrating the MDA-MB-231 without CAF conditions come from the same videos analyzed for Fig 2 but with different time-interval resolution. For statistical analysis, the measurements for each of the 4 conditions were assembled regardless of the time variable (n = 51 per condition). The different CAF/drug conditions were compared to the MDA-MB-231 control condition. The data passed the Shapiro-Wilk normality test, therefore the unpaired t test was applied. ** indicates p<0.001, * indicates p<0.05 and ns indicates not significant.

We computed $P_{death}$ for the videos of both breast MDA-MB-231 and lung IGR-Pub cells (Figs 6 and S2). In basal conditions, without drug or T cells, $P_{death}$ is low for both cell types (< 0.1–0.2 x10$^{-3}$) and globally stable over the experimental time (2–3 days), meaning that naturally dying cancer cells do not have an impact on viability of nearby cells. When cytotoxic death of MDA-MB-231 cells is induced with doxorubicin, $P_{death}$ gradually increases up to 3–4 folds during the first 2 days, and remains high during the 3$^{rd}$ day, meaning that doxorubicin-dependent cytotoxic death actually promotes the death of nearby cells. Similarly, when death of IGR-Pub cells was induced by autologous CTL, $P_{death}$ is higher than the control without T cells from the start of co-cultures, and further increases during the 2-day experimental time, suggesting that T-cell-dependent cytotoxic death promotes the death of nearby cells as well. When MDA-MB-231 cells are co-cultured with CAFs, the $P_{death}$ remains low without and with doxorubicin (Fig 6C), providing another evidence of the CAF-dependent chemo-resistance.

$P_{death}$ increase does not simply result from the increase of death numbers over time, but it depends also on the death positions, as shown by the simulations reported in S3 Fig. Indeed, an artificial video, with the same rate of death events as a real video (in this case, MDA-MB-231 cells treated with 1 μM doxorubicin), but with a spatially random distribution which maintains approximatively the relative death object distances as the real video, displays a computed $P_{death}$ increase much lower than the one measured for the real video. Moreover, when artificial videos with the same rate of death events but different spatial distribution (random versus clustered) are compared, the computed $P_{death}$ is much higher for the clustered deaths than for the random deaths, but in both cases it does not increase over time.

Therefore, the real $P_{death}$ kinetics depend on both temporal and spatial features, and suggest the possibility that, contrary to naturally dying cancer cells, the cells that enter into apoptosis triggered by chemotherapy or T cells, send pro-apoptotic signals to neighbor cells, initiating a chain of death that amplifies the initial cytotoxic effect.

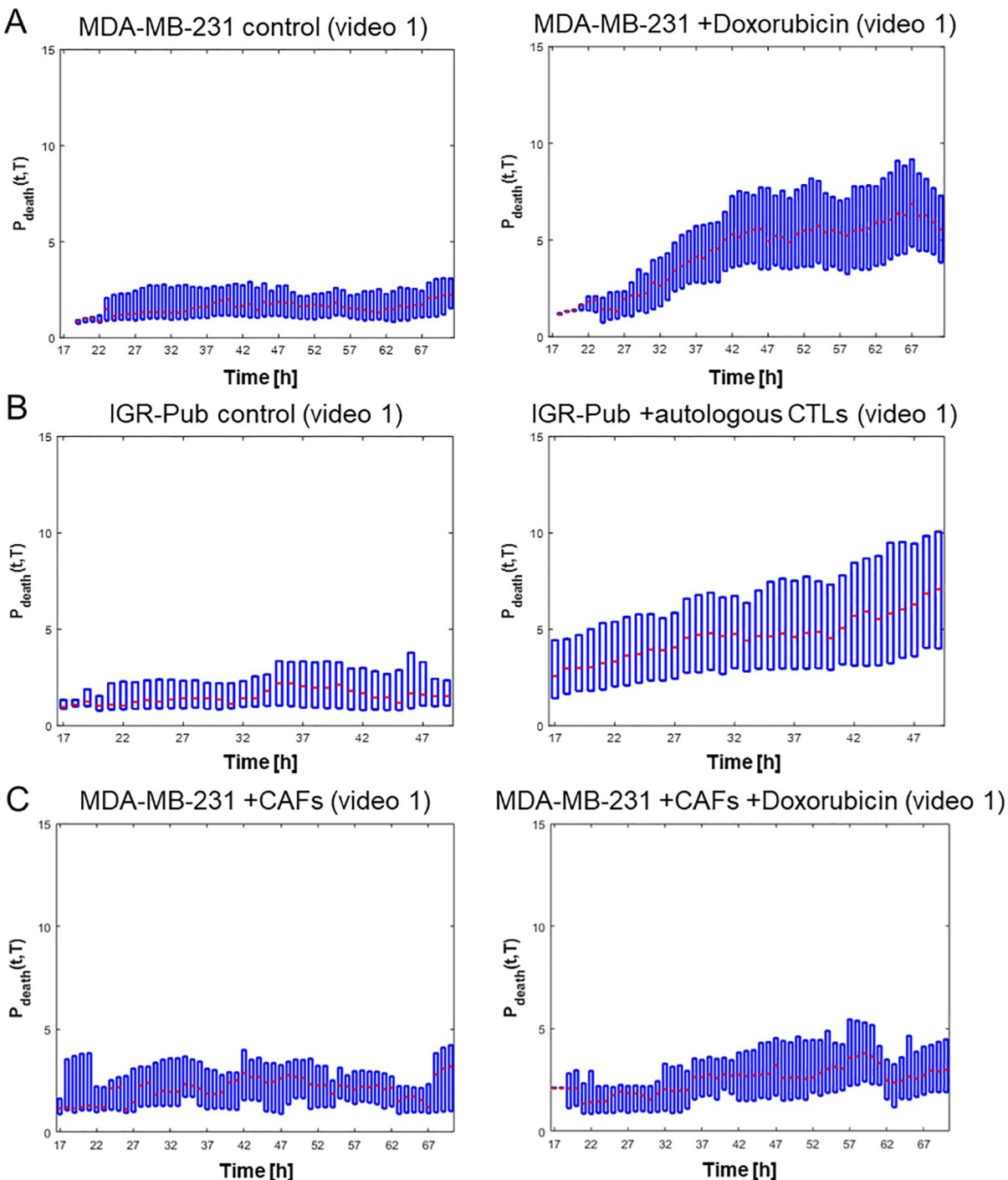

**Fig 6. The Potential of death induction ($P_{death}$) of cancer cells increases over time upon cytotoxic death, but not upon natural death. A.** Representative $P_{death}$ analysis on one video of breast cancer MDA-MB-231 cells cultured within ToC without (left, natural death) or with 1 μM doxorubicin (right, cytotoxic death). **B.** Representative $P_{death}$ analysis on one video of NSCLC IGR-Pub cells cultured within ToC alone (left, natural death) or together with autologous CTL (P62 clone) (right, cytotoxic death). **C.** Representative $P_{death}$ analysis on one video of breast cancer MDA-MB-231 cells co-cultured with CAFs within ToC, without (left) or with 1 μM doxorubicin (right). More video analyses on additional view fields are shown in S2 Fig.

## Discussion

### The STAMP method

We report here a new method, STAMP, which extracts the temporal kinetics and the spatial maps of cancer death events within ToC co-cultures. The robustness and versatility of the method is demonstrated by its successful application to different cell models (breast and lung cancers), co-culture combinations (cancer cell alone, or together with T cells or CAFs), and experimental time-lapse acquisitions (frequency and duration). In principle, the STAMP method could be used to measure the death of other cells types within ToC, such as immune cells or fibroblasts, by selectively pre-staining one of these populations. However, some adjustments will be required. First, the time acquisition rate needs to be adapted to the motility of the specific cell type; for example, T-cell tracking would require a temporal resolution much higher than cancer cells, with time intervals less than 1 min [24]. Second, the STAMP code needs to be adapted to different cell morphologies; for example, the cell radius parameter must be revised.

Moreover, the STAMP image analysis method might be useful for many other cellular contexts and biological questions, beyond the ToC technology. This adaptability to multiple experimental conditions is possible thanks to the integration within the STAMP software of three modular parameters.

First, the $T_{\mathrm{LAG}}$ mentioned in step 6-iii and used in Eq (4) (see Materials and Methods). $T_{\mathrm{LAG}}$ is the temporal window used to measure the average number of apoptotic events $N_{ap}(t, T_{LAG})$ and the average number of living cells $N_{avg}(t, T_{LAG})$, allowing to compute the percentage of apoptosis events. $T_{\mathrm{LAG}}$ should be set depending on the time frame image acquisition frequency and on the desired time resolution of the investigated phenomenon. For example, the acquisition frequency being every 1 h in this study, the choice of $T_{\mathrm{LAG}}$ value is bounded below. However, $T_{\mathrm{LAG}}$ values larger than acquisition frequencies were more appropriate to avoid spurious fluctuations due to uncontrollable changes affecting the measurements (*e.g.*, abrupt change in illumination). Conversely, $T_{\mathrm{LAG}}$ is also bounded above by the necessity to avoid flattening dynamic phenomena. We set $T_{\mathrm{LAG}}$ to 10 h for Figs 2 and 3 whose purpose was the compare global accuracies, and to 4 h for Fig 4 whose purpose was to compare kinetics.

Second, the parameter $r$, included in the morphological operator $\Psi_{\mathrm{E}}^{\mathrm{B}}$ (Eq (7)). The dimension of $r$ impacts on the death wake construction. In particular, starting from a circular object of radius $r_{tc}$ the effect of the application of the operator in Eq (7) is to restrict the object radius of a quantity equal to $r$. Hence, by indicating with $t_0$ the time at which the cell $tc$ dies and with t a generic time frame such that $t > t_0$, the radius vanishes according to the formula

$$r_{tc}(t) = \max\left(0, r_{tc}(t_0) - r \cdot (t - t_0)\right)$$

By setting $r$ as one third of $r_{tc}$ the equation can be re-written as

$$r_{tc}(t) = \max\left(0, r_{tc}(t_0) - \frac{1}{3}r_{tc}(t_0) \cdot (t - t_0)\right) =$$

$$= \max\left(0, r_{tc}(t_0)\left(1 - \frac{1}{3}(t - t_0)\right)\right) = \max\left(0, r_{tc}(t_0)\left(1 - \frac{1}{3}\Delta t\right)\right)$$

where it can be noted that for at least three hours ($\Delta t = 3$) the wake exists.

Third, the parameter $\tilde{T}$, in computation of MC (Eq (9)) and of $P_{death}$ (Eq (11)). $\tilde{T}$ is the time window over which the aggregation of deaths and their wake were computed by means of the definition of the cumulative map $MC$ (Eq (9)). Then, for each $t \in \{1, ..T - \tilde{T}\}$, the potential of death induction $P_{death}(t, \tilde{T})$ simultaneously measured the spatial and temporal death

induction effects at time $t$. The value of $\tilde{T}$ has a key role in the quantification of death induction: a too small $\tilde{T}$ value results in the under-detection of genuine death induction effects, while a too large $\tilde{T}$ value causes a misleading flattening effect.

In this study, we set $\tilde{T} = 16$ h for both breast and lung cancer cells, based on the mathematical investigation of an *induction interval* that was associated to each cell and computed as follows. Let us consider a tumor cell $tc$ centered in $(x_{tc}(t), y_{tc}(t))$ with radius $r_{tc}(t)$ at the time of its death, $t = T_{tc}^{death}$. We assume that from its beginning at $t = T_{tc}^{death}$, the apoptosis of the cell $tc$ induces some deaths and these deaths cause others and so on, by creating a *chain of death* started from the cell $tc$. We constructed this chain by involving deaths that occurred in a circular zone of radius equals to $10 \cdot r_{tc}(t)$, centered in $(x_{tc}(t), y_{tc}(t))$, at a temporal distance from each other equals to $T_{LAG}$. The total duration of the *chain of death* defines the *induction interval* related to the cell $tc$. The distributions of the duration of the induction intervals of cells from 16 videos from 2 experiments (S4 Fig), show that vast majority of induction intervals is below 16 h, meaning that $\tilde{T} = 16$ permits to capture the vast majority of chain of death events.

## The biological implications

Among the various types of cell death [25], this work specifically investigates the programmed apoptosis which involves the activation of the cascade of caspase enzymes. Both cytotoxic stimuli we used are known to promote apoptosis of cancer cells. Doxorubicin, which acts by causing irremediable DNA damages in dividing cells, has been shown to induce apoptosis on breast cancer cell lines *in vitro* [26], as well in breast cancer patients *in vivo* [27]. The CTL clone (P62) used in this work kills autologous cells (IGR-Pub) at least in part via Apo2L/TRAIL-dependent pathway [15].

By using novel mathematical (potential of death) and computational (STAMP software) strategies, we achieved an original spatiotemporal analysis of apoptotic cancer death in the 3D confined environments of ToC cultures. Surprisingly, contrary to natural death, both doxorubicin-dependent and T-cell-dependent cytotoxicity toward target cells promoted the death of nearby cancer cells, indicating that dying cancer cells might release soluble pro-apoptotic signaling factors and trigger a chain of death that amplifies the initial cytotoxic stimulus.

Apoptotic cells do not passively empty their cellular content but they actively release various signals, named as damage-associated molecular-pattern (DAMP) molecules [28]. First, they release 'find-me' and 'eat-me' signals (such ATP and UTP nucleotides, the CX3CL1 chemokine, and the bioactive lipid metabolites phosphatidylserine (PS), lysophosphatidylcholine (LysoPC) and sphingosine-1-phosphate (S1P)), which enhance the attraction of phagocytes to dying cells and their consequent phagocytic clearance (a process called efferocytosis) [29]. Second, they send metabolite 'good-bye' signals with biological functions (such as AMP, GMP, creatine, spermidine, glycerol-3-phosphate (G3P), ATP), which act as tissue messengers altering gene expression of healthy nearby cells, for example suppressing inflammation [30]. Third, in the case of apoptotic cancer cells, they secrete cytokines/chemokines (such as IL-8, CCL2, CXCL1, CXCL2, CXCL5) that act as 'immunomodulatory' signals, promoting for example the polarization of monocytes to M2-like cells with consequent establishment of a tumor-supportive immune microenvironment [31]. Fourth, our findings indicate that apoptotic cancer cells release unknown 'pro-apoptotic' signals that directly induce death of neighboring cells, without intervention of phagocytic macrophages, absent in our co-culture models. Identification of these compounds warrants more work. We estimated that the diffusion times of potential signaling molecules of various sizes (*e.g.*, 100–500 Da for nucleotides, lipid metabolites, organic compounds or 10–20 kDa for cytokines), between cells with a distance in the 20–100 micron range, within the collagen gel (2.3 mg/mL), are very low, less than 10 minutes. Therefore, these

diffusion times are fully compatible with the hypothesis of release of pro-apoptotic compounds from dying cells, triggering chains of deaths that last for 2–14 hours (S4 Fig). Importantly, the biology of development teaches us that many secreted factors control apoptotic death events, spatially and temporally, to build multicellular organisms [32]. Pro-apoptotic compound candidates might be searched among these already known killers. However, at this stage we cannot exclude the implication of other processes in shaping the cytotoxic spatiotemporal behaviors. For example, the 20 h latency in the response to doxorubicin (Fig 2C, right) might be caused by the necessity for cancer cells to enter DNA replication phase or by the time-dependent intracellular accumulation of this drug [33]. In lung cancer-T cell co-cultures, the late burst of death after 30 h of co-culture (Fig 3C, right) might be due at least in part to a further activation of T cells upon co-culture with the target cells.

The phenomenon of 'death transmissibility' or 'death contagiousness' we unveiled in our *ex vivo* experimental setting might be linked to the well-known bystander effects observed in clinics [34]. Radiation-induced or chemotherapy-induced or immunotherapy-induced bystander effects refer to the induction of biological effects in cells that are not directly treated by radiation or chemotherapy or immunotherapy, but are in close proximity to cells that are. In our specific cases, all cancer cells are treated with doxorubicin or co-cultured with CTL, but the cells for which the treatments are effective have an indirect, unexpected, effect on the nearby cells.

In conclusion, this interdisciplinary work, by combining cancer biology, microfluidic engineering, mathematical modeling and computational analysis, created an innovative and needed image analysis method (STAMP), confirmed the power of ToC technology and shed a new light on the complexity of tumor ecosystem, emphasizing the intricacy of its non-autonomous cell behaviors.

## Materials and methods

### Cell cultures

The MDA-MB-231 cell line, from triple negative breast cancer, was cultured in high-glucose DMEM (GE Healthcare, #SH30081.01) supplemented with 10% fetal bovine serum (Biosera), 1% Penicillin/Streptomycin (Gibco), 1% glutamine (Gibco). The IGR-Pub lung adenocarcinoma cells and the autologous T cells P62 were generated from the same patient in one of our laboratories at Institut Gustave Roussy [15]. The IGR-Pub cells were cultured in DMEM F12 (GIBCO) supplemented with 10% fetal bovine serum (Biosera), 1% of Ultroser G (Pall), 1% of Sodium Pyruvate (Gibco) and 1% Penicillin/Streptomycin (GIBCO). P62 T cells were cultured in RPMI-1640 (GE Healthcare) supplemented with 10% human AB serum (Institut Jacques Boy, Reims, France), rIL-2 (20 U/ml, Gibco), 1% of Sodium Pyruvate (Gibco) and 0,1% Penicillin/Streptomycin (Gibco). Primary cancer-associated fibroblasts (CAFs) were isolated and cultured as previously reported [8,23]. All cell lines were periodically tested to exclude mycoplasma contamination using a qPCR-based method (VenorGem Classic, BioValley, #11–1250). The MDA-MB-231 cell line was authenticated by short tandem repeat (STR) profiling (GenePrint 10 system, Promega, #B9510). Doxorubicin was purchased from Teva pharmaceuticals (200 mg/100 ml).

### Tumor-on-chip preparation

The microfluidic devices were purchased from AIM-Biotech (#DAX-1). Cells were seeded in the central chamber of the DAX-1 chips embedded in a matrix composed of type I rat tail collagen (Thermofisher, #A1048301) at the final concentration of 2.3 mg/ml. Cancer cells were seeded in the gel at a final density of $2x10^6$ cells/ml. Autologous T cells were added at final

densities of $0.2 \times 10^6$ to $2 \times 10^6$ cells/ml in order to obtain different ratios (from 10:1 to 1:1) between cancer and T cells. Primary CAFs were added at CAF:cancer 1:6 ratio. The microfluidic devices were incubated for 30 min at 37°C in a humidified chamber to allow the polymerization of the collagen solution; afterwards, 120 μl of culture medium were added in each lateral chamber. MDA-MB-231 cells in chip were cultured in the same medium used for dish 2D culture, whereas the IGR-Pub/P62 co-cultures were in T-cell medium, supplemented with rIL-2 (10 U/ml, GIBCO, #PHC0027). After the addition of the medium, the microfluidic devices were kept for 1 h in the incubator before transfer to the incubating chamber of the microscope for imaging.

## Cell staining

Cancer cells were labeled with CellTrace Yellow before seeding in the gel (Thermofisher, #C34567), for the detection in the so-called "red channel" of fluorescence: cells were trypsinized, and then resuspended at $1 \times 10^6$ cells/ml density in PBS with 5 μM CellTraceYellow; after incubation in cell medium for 5 min at 37°C, cells were centrifuged at 300g for 5 min, resuspended in PBS and added to the rat-tail collagen solution.

CellEvent Caspase-3/7 Green Detection Reagent (Thermofisher, #C10423) was added to the medium in the lateral chamber of the chip in order to visualize in the "green channel" the cells undergoing apoptosis.

## Live cell imaging

Time-lapse images were acquired with an inverted Leica DMi8 equipped with a Retiga R6 camera and Lumencor SOLA SE 365 light engine, using a 5X objective. The filter cubes used were TXRed (excitation filter 560/40 nm, emission filter 630/75 nm, dichroic mirror 585 nm) and GFP (excitation filter 470/40 nm, emission filter 525/50 nm, dichroic mirror 495 nm). The exposure times were 20 ms for the bright field and 700 ms for the fluorescent channels. The video-microscope was equipped with a motorized stage for multi-positioning acquisition, a $CO_2$ and temperature-controlled (37°C) incubator chamber (S5A Fig). The presence of a saturating humidity in the microscope chamber is crucial for optimal cell viability, therefore distilled water was added in the plastic wells of the DAX-1 chips and humidified small sponges were added in the chip surroundings (S5B Fig). Since in the AIM-Biotech devices the gas-permeability is provided by the underside sealing layer, before inserting them on the microscope stage, we placed them on a standard microscope glass slides and we lifted them using magnet holders (1 mm thick), in order to create an air circulating space underneath the devices, for $CO_2$ and temperature control (S5C and S5D Fig). The acquisition of images in transmission and fluorescent channels was performed every hour for a total duration of 48 h to 72 h.

## The STAMP method

A STAMP software was developed under MATLAB environment and can be downloaded using the following link: https://cloudstore.bee.uniroma2.it/index.php/s/LEpHYTsPnDj4Ajt (password: STAMP2021).

The STAMP method was applied on each video $V$, with spatial dimensions $D_1$ (number of row) and $D_2$ (number of columns) and with a total duration of $T$ frames (from 48 to 72 depending on experiments, with a frame rate of 1 h).

Let us consider $(x,y,t) \in \{1,..,D_1\} \times \{1,..D_2\} \times \{1,..T\}$ the tuple indicating the position of the coordinates $(x,y)$ occupied by an arbitrary pixel on the video frame of $V$ acquired at time $t$, where $t = 1,...,T$. Then, $V(x,y,t)$ indicates the video sequence with the specific coordinates $(x,$

$y$) at time $t$. We can refer to the video under examination indistinctly with $V$ or $V(x,y,t)$ to vary of $(x,y,t) \in \{1,..,D_1\} \times \{1,..D_2\} \times \{1,..T\}$.

1. **Cell localization and tracking.** Tumor cells (stained in red) were located and tracked in the red channel video of $V$ by adapting Cell-Hunter software to the frame rate of 1 h [7,35,36]. Localization was performed by preliminary binarizing the red channel video of $V$ by Otsu approach [37]. Then, Cell-Hunter was applied. Shortly, in each binarized video frame, the software implements the Circular Hough Transform (CHT) [38] which automatically locates tumor cells, assumed as circular-shaped objects of radius imposed, providing an accurate estimate of individual cell radii. Then cell trajectories/tracks were constructed by linking positions between consecutive frames according to an optimized procedure based on the concept of cell proximity and optimal assignment problem [39].

2. **ROI extraction around each tumor cell.** After tracking all the tumor cells along the video $V$, we isolated a square region of interest (ROI) 31 pixels x 31 pixels (about 20μm), centered around each tumor cell position along each track. In this way, we constructed a square section tube around each track. This procedure allowed us to confine the next analysis in the neighborhood of the tumor cells and to limit confounding factors in apoptosis analysis due to surrounding cells.

3. **Background and foreground identification.** Each ROI visualizes the cell (the foreground) and the background culture environment. To separate them, we segmented the tumor cells in the ROI by CHT approach and determined a neighborhood circular region around the cell by a given radius, here set to double the average radius of tumor cells.

4. **Time-dependent green emission signal extraction.** In order to extract the green emission signals of tumor cells (*i.e.* tumor apoptosis events), we transposed from the red to the green channel video the tracked positions of tumor cells (*i.e.* the centers of the cell regions automatically detected by Cell-Hunter software).

Let us denote as $(x_{tc}(t), y_{tc}(t)) \in \{1,..,D_1\} \times \{1,..D_2\}$ the pixel representing the position of the arbitrary tumor cell $tc$ at time-frame $t$ in the red and then green channel of video $V$, with $t \in F \subseteq \{1,..,T\}$, where $F = \{t_{tc}^{start}, \ldots, t_{tc}^{end}\}$ is the set of time-frames for which the cell track constructed by Cell-Hunter exists. We can define:

- $I_{GREEN}(x,y,t)$, the intensity value of the green emission signal referred to the pixel in position $(x,y)$ on the video frame acquired at time $t$ in the green channel, *i.e.*, the green channel video sequence of $V(x,y,t)$, with $(x,y,t) \in \{1,..,D_1\} \times \{1,..D_2\} \times \{1,..,T\}$,

- $R(x_{tc}(t), y_{tc}(t))$, the squared ROI centered on the position $(x_{tc}(t), y_{tc}(t))$ of the tumor cell $tc$ at time $t$, $t \in F$,

- $R_B(x_{tc}(t), y_{tc}(t))$ and $R_F(x_{tc}(t), y_{tc}(t))$, the circular background and the circular foreground regions within the ROI $R$, respectively, both centered on the position of the same tumor cell $tc$ at the same time $t$, $t \in F$.

Moreover, if $\bar{R}(x_{tc}(t), y_{tc}(t))$ is a generic ROI centered on $(x_{tc}(t), y_{tc}(t))$, *i.e.*, $\bar{R} = R \vee R_B \vee R_F$ for $t \in F$, and $(x,y)$ is a pixel on the video frame acquired at time $t$ in the green channel belonging to the ROI $\bar{R}(x_{tc}(t), y_{tc}(t))$, we can write $(x,y) \epsilon \bar{R}(x_{tc}(t), y_{tc}(t))$.

Then, in order to capture the information content of the green emission in the tumor cell $tc$ at a time $t \in F$, we proceed as follows

**4-i** Compute the average green emission signal in the foreground region $R_F(x_{tc}(t),y_{tc}(t))$, expressed by

$$\mu_{tc}^{R_F}(t) = \frac{1}{Area(R_F(x_{tc}(t), y_{tc}(t)))} \sum_{(x,y) \in R_F(x_{tc}(t),y_{tc}(t))} I_{GREEN}(x, y, t). \quad (1)$$

**4-ii** Compute the average green emission signal in the local background $R_B(x_{tc}(t),y_{tc}(t))$, that is

$$\mu_{tc}^{R_B}(t) = \frac{1}{Area(R_B(x_{tc}(t), y_{tc}(t)))} \sum_{(x,y) \in R_B(x_{tc}(t),y_{tc}(t))} I_{GREEN}(x, y, t). \quad (2)$$

**4-iii** Perform background subtraction and normalization correction in order to avoid misleading surrounding green emission, thus obtaining $\mu_{tc}(t)$ as follows

$$\mu_{tc}(t) = \frac{\mu_{tc}^{R_F}(t) - \mu_{tc}^{R_B}(t)}{\mu_{tc}^{R_B}(t)} - \min_{\hat{t} \in F} \left( \frac{\mu_{tc}^{R_F}(\hat{t}) - \mu_{tc}^{R_B}(\hat{t})}{\mu_{tc}^{R_B}(\hat{t})} \right). \quad (3)$$

By computing $\mu_{tc}(t)$ for each $t \in F$, the time-depending signal $\mu_{tc}$ referred to the track of the tumor cell $tc$ is produced. The higher the signal is the higher is the green emission of the cell region and the probability to have an apoptosis event.

5. **Detection of the beginning of the apoptosis events.** Let assume $N$, the total number of detected tumor cells along the entire duration of the video $V$. From the previous step, $N$ time-dependent signals $\mu_{tc}$ were computed, one for each of tumor cells denoted as $tc$. We estimated a threshold value $th$ as the optimal inter-variance separation value of all the $N$ signals $\mu_{tc}$ (Otsu approach) [37]. For each tumor cell $tc$, we considered that the death by apoptosis occurs if $\mu_{tc} > th$ and that apoptosis begins at the time-frame at which the $\mu_C$ exceeds the value $th$ for the first time, $T_{tc}^{death} = \min_t\{t \in F | \mu_{tc}(t) > th\}$.

6. **Counting the apoptotic events.** In order to count the apoptotic events, we have to move from a tumor cell-centric view, used in depicting Steps 2–5, to a time-centric view. So, for each $t \in \{1,...,T\}$, we followed the approach below:

**6-i** Compute the number of apoptosis at time-frame $t$, $N_{ap}(t, T_{LAG})$, which sum up the number of apoptosis events found in the range $[t-T_{LAG}, t]$ as the cumulative number of tracks of tumor cells $tc$ whose signal $\mu_{tc}$ satisfies the condition $\mu_{tc}(t) > th$, for all $t \in [t-T_{LAG}, t]$.

**6-ii** Compute the number of tracks of living cells at time-frame $t$, $N_{track}(t)$, as the number of tracks at time $t$ that did not yet go into apoptosis, *i.e.* the number of tracks whose $\mu_{tc}$ at time $t$ satisfies the condition $\mu_{tc}(t) < th$.

**6-iii** Compute the average number of tracks found in a temporal lag of $T_{LAG}$ frames, $N_{avg}(t, T_{LAG})$, as the average of $N_{track}(t)$ in the range $[t-T_{LAG}, t]$. The value of $T_{LAG}$ was defined in the order of a few hours (2–10 h) according to the desired temporal resolution and heuristic investigation (see Discussion).

**6-iv** Compute the percentage of apoptotic events in $T_{LAG}$ frames, $O(t, T_{LAG})$, as

$$O(t, T_{LAG}) = \frac{N_{ap}(t, T_{LAG})}{N_{avg}(t, T_{LAG})} \cdot 100\% \quad (4)$$

**6-v** Compute the average of surviving cells in each time point $N_{avg2}(t, t_{\pm 1})$, between 3 time points centered on t, as the average of $N_{track}(t)$ in the range $t_{\pm 1} = [t_{-1} - t_{+1}]$.

**6-vi** Compute the percentage of surviving cells, $OS(t, t_{\pm1})$ (also referred as "overall survival"), as

$$OS(t, t_{\pm1}) = \frac{N_{avg2}(t, t_{\pm1})}{N_{avg2}(t, t_1 - t_3)} \cdot 100\% \tag{5}$$

Where $t_1$ and $t_3$ are the first and third time points.

7. **Construction of spatiotemporal maps of apoptotic events.** By using the information of death of the single tumor cell $tc$ (namely, position, $(x_{tc}(t), y_{tc}(t))$ for each $t \in F$ and timing of the apoptotic event, $T_{tc}^{death}$), we constructed a spatiotemporal map of death by the following procedure:

**7-i** An artificial video with the same spatial and temporal dimensions of video $V$, ($D_1$, $D_2$, $T$, respectively), was generated such that, for each tumor cell $tc$, at frame $t = T_{tc}^{death}$, the cell region, assumed as a circle and centered in the position $(x_{tc}(T_{tc}^{death}), y_{tc}(T_{tc}^{death}))$, was labelled with a white pixels, *i.e.*, with pixel intensity values equal to 1. It allowed to artificially reproduce the cell region of each tumor cell $tc$ at its time of death, $T_{tc}^{death}$.

Let us indicate with $MD(x, y, t)$ the artificial video sequence, with $(x, y, t) \in \{1, .., D_1\} \times \{1, .. D_2\} \times \{1, .. T\}$.

**7-ii** By assuming a spatiotemporal signaling of death produced by cells going into apoptosis, we constructed a new artificial video, $M(x, y, t)$, according to an iterative approach, expressed by:

$$M(x, y, t) = \Psi_E^B(M(x, y, t - 1)) + MD(x, y, t), \tag{6}$$

with $(x, y, t) \in \{1, .., D_1\} \times \{1, .. D_2\} \times \{2, .. T\}$, and $M(x, y, 1) = MD(x, y, 1)$.

The operator $\Psi_E^B$ denotes the gray-scale morphological erosion operator [36], with structure element $B$, defined as:

$$\Psi_E^B((M(x, y, t)) \triangleq \min_{(x', y') \in B} \{M(x + x', y + y', t)\}. \tag{7}$$

It is the extended binary erosion operator defined on gray-scale intensity matrices. The global effect of the $\Psi_E^B$ operator is to reduce the area occupied by each white objects in the processed frame thus implementing a vanishing signaling that we called a *death wake* (see death signaling modeling step in Fig 1).

To apply the operator $\Psi_E^B$, in the present work, we used a circular structure element with radius $r$, *i.e.*, $B_r$ defined as:

$$B_r \triangleq \{(x, y) \mid x^2 + y^2 \leq r\} \tag{8}$$

where the parameter $r$ is defined as one third of the estimated average cell radius in the experiment. The choice of $r$ depends on the need to simulate a wake with a reasonable duration with respect to the timing of the experiments (see Discussion).

The constructed artificial video $M(x, y, t)$ takes into account the death wakes of cells enabling to cumulate the death signaling in a given region.

7-**iii** We combined in a unique index both spatial and temporal death influence. First, given a temporal windowing of size $\tilde{T}$, we designed the cumulative map $MC$ defined as:

$$MC(x, y, t, \tilde{T}) = \sqrt{\sum_{t'=t}^{t+\tilde{T}} (M(x, y, t'))^2}, \tag{9}$$

with $(x, y, t) \in \{1, .., D_1\} \times \{1, ..D_2\} \times \{1, ..T - \tilde{T}\}$. The cumulative map allows to aggregate the death events and their wakes over a given temporal interval equal to $\tilde{T}$, whose value needs to be optimized (see Discussion). Then, to account for spatial influence (*i.e.*, to discriminate randomly versus deterministically spatially distributed deaths) we defined a *potential of death induction* (S1 Fig).

Let us consider the temporal map $(x, y, t, \tilde{T})$. Since cell death is a sparse phenomenon, most part of the map $MC$ is null. Hence, we can define a generic object $s(t)$ at frame $t$ as a region of the map $MC$ at time $t$ that is not connected with other non-null region, and indicate with $S(t)$ the set of not connected objects,

$$S(t) \triangleq \{s_i(t) | s_i(t) \cap s_j(t) = \emptyset, i \neq j\}. \tag{10}$$

Connection is defined under the 8-connectivity criterion [36]. Under these assumptions, for each $t \in \{1, ..T - \tilde{T}\}$, we defined the potential of death induction as follow:

$$P_{death}(t, \tilde{T}) \triangleq \frac{1}{2|S(t)|} \sum_{i=1}^{|S(t)|} \sum_{j=i+1}^{|S(t)|} \frac{\text{mean}_{(x,y) \in si(t)} MC(x, y, t, \tilde{T}) + \text{mean}_{(x,y) \in sj(t)} MC(x, y, t, \tilde{T})}{\bar{d}(s_i(t), s_j(t))}, i \neq j \tag{11}$$

where $|S(t)|$ denotes the number of elements in $S$ and $\bar{d}(s_i(t), s_j(t))$ denotes any distance operator between objects $s_i(t)$ and $s_j(t)$, normalized by the maximum dimension of the video frame. In this work, we chose the Euclidean distance between the geometrical center of the two objects, *i.e.*, the average coordinates of their boundary.

In the computation of the $P_{death}$, Eq (11) is repeatedly applied to a sliding detection window of dimension 283 μm x 283 μm to cover the entire view field. In this way, not only the death signal is spatially confined, but also the statistical relevance of the calculus is increased at each time point. We define as 'nearby cells' those falling in this 283 μm x 283 μm window.

## Statistical analysis

Statistical analysis and graphs were made with GraphPad Prism software (v8). We first performed a Shapiro-Wilk normality test. When the conditions passed the normality test, we applied a parametric t test. If not, we performed the non-parametric Mann-Whitney test. Statistical threshold for significance was set for p-values inferior to 0.05.

## Supporting information

**S1 Fig. Simulated example for the calculation of the Potential of death induction ($P_{death}$).** Three simulated deaths occur at 2 h, 9 h, and 11 h (first video sequence). The death signals are modeled by the construction of a signal wake (second video sequence), the duration of which depends on the dimension of the original death region (first video sequence). Then, a cumulative map $MC(x, y, t, \tilde{T})$ is constructed by combining both spatial and temporal death influence (third video sequence) using $\tilde{T} = 6$. Finally, $P_{death}$ is computed over time for the entire image area (bottom graph). Until t = 8 h, there is only one death, so there is no induction phenomenon. An additional death occurs at t = 9 h thus producing an induction phenomenon and an increase in potential. A third death occurs at t = 11 h thus producing a further increase in the potential value. Potential is also influenced by the absolute value of the map $MC$ and by the distances of the different zones of death. From t = 12 h there is no more memory of the first death, hence only the last two death zones remain whose distance is larger than that of the two death zones involved in t = 9 h and 11 h, thus causing a decrease in potential.
(PNG)

**S2 Fig. Addition $P_{death}$ analyses supporting Fig 6.**
(PNG)

**S3 Fig. Simulations showing the dependency of the Potential of death induction ($P_{death}$) on temporal and spatial features. A.** Experimental data showing the spatial localization of death events at different time points (above), and the corresponding $P_{death}$ measurements (below) on a video of MDA-MB-231 cells treated with 1 μM doxorubicin (the same reported in Fig 6A). **B.** Simulation of a video with the same death events as in A, but with a spatially random distribution, maintaining approximately the relative object distances. Note that the corresponding $P_{death}$ measurements are increasing much less than in A, indicating that the $P_{death}$ increase does not simply result from the increase of death numbers over time, but it depends also on death positions. **C.** Simulation of a video with a constant number of death events with a spatially random distribution. Note that the corresponding $P_{death}$ measurements are constant over time. **D.** Simulation of a video with a constant number of death events with a clustered distribution. Note that the corresponding $P_{death}$ measurements are constant over time, but higher than in C.
(PNG)

**S4 Fig. Rationale for the calculation of the optimal $\tilde{T}$.** $\tilde{T}$ is the time window over which the aggregation of deaths and their wake were computed by means of the definition of the cumulative map $MC$ (Eq (9)). The induction intervals, defined as the duration of the chain of death, were computed for each cell, from 16 videos from 2 experiments, one experiment with the lung cancer cell line IGR-Pub (A) and one experiment with the breast cancer cell line MDA-MB-231 (B). The distributions of induction show that vast majority of induction intervals is below 16 h, meaning that $\tilde{T} = 16$ h is an optimal choice.
(PNG)

**S5 Fig. Images showing the microscope setup for ToC cultures. A.** Global view of the Leica DMi8 used to perform the live imaging experiments. The white arrow points at the heating black chamber, in which is maintained a temperature of 37°C. The temperature is set and maintained by a temperature controller (blue arrow). The $CO_2$ controller (black arrow) mixes $CO_2$ with atmospheric air. Through a tubing system, the air with the controlled $CO_2$ at 5%, after passing through a bottle half filled with water for humidification (red arrow), is injected in the microscope chamber. **B.** View of the chip placed in the microscope chamber. The white arrow indicates the lid of the chamber in which is injected the humidified air with $CO_2$ at 5%. The red arrow points at the chip to be imaged. Humidified sponges contribute to humidify the chamber and to minimize micro-evaporation phenomena (black arrows). **C.** Picture from the side of the chip filled with medium. At each 'anchor' point, three magnets (red arrows) are used to lift and attach the chip to the glass slide (black arrow). **D.** Top view of an empty chip with magnets. Two piles composed of three magnets (red arrows) are applied in the central part of the chip and glass slide.
(PNG)

**S1 Movie. Breast cancer MDA-MB-231 cells alone.**
(AVI)

**S2 Movie. Breast cancer MDA-MB-231 cells with doxorubicin.**
(AVI)

**S3 Movie. Lung cancer IGR-Pub cells alone.**
(AVI)

**S4 Movie. Lung cancer IGR-Pub cells with autologous T cells (ratio 1 to1).**
(AVI)

**S5 Movie. Breast cancer MDA-MB-231 cells with CAFs.**
(AVI)

**S6 Movie. Breast cancer MDA-MB-231 cells with CAFs with doxorubicin.**
(AVI)

**S7 Movie. Representative STAMP output video.**
(AVI)

## Acknowledgments

We are grateful to Ariel Savina for inspiring discussions at the beginning of the project, and to Elodie Voilin and Jamila Kacher (U1186, IGR) for T-cell clone amplification.

## Author Contributions

**Conceptualization:** Irina Veith, Arianna Mencattini, Jacques Camonis, Hamasseh Shirvani, Fatima Mechta-Grigoriou, Gérard Zalcman, Maria Carla Parrini, Eugenio Martinelli.

**Formal analysis:** Irina Veith, Arianna Mencattini.

**Funding acquisition:** Fatima Mechta-Grigoriou, Gérard Zalcman, Maria Carla Parrini, Eugenio Martinelli.

**Investigation:** Irina Veith, Arianna Mencattini, Valentin Picant, Maria Colomba Comes.

**Methodology:** Marco Serra, Stéphanie Descroix.

**Resources:** Marine Leclerc, Fathia Mami-Chouaib.

**Software:** Arianna Mencattini, Eugenio Martinelli.

**Supervision:** Hamasseh Shirvani, Maria Carla Parrini, Eugenio Martinelli.

**Writing – original draft:** Irina Veith, Arianna Mencattini, Maria Carla Parrini, Eugenio Martinelli.

**Writing – review & editing:** Irina Veith, Arianna Mencattini, Valentin Picant, Marco Serra, Marine Leclerc, Maria Colomba Comes, Fathia Mami-Chouaib, Jacques Camonis, Stéphanie Descroix, Hamasseh Shirvani, Fatima Mechta-Grigoriou, Gérard Zalcman, Maria Carla Parrini, Eugenio Martinelli.

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
