## [Decision Letter · Decision Letter 0]

25 Nov 2020

Dear Prof. Martinelli,

Thank you very much for submitting your manuscript "Apoptosis mapping in space and time of 3D tumor ecosystems reveals transmissibility of cytotoxic cancer death" for consideration at PLOS Computational Biology.

As with all papers reviewed by the journal, your manuscript was reviewed by members of the editorial board and by several independent reviewers. In light of the reviews (below this email), we would like to invite the resubmission of a significantly-revised version that takes into account the reviewers' comments.

We cannot make any decision about publication until we have seen the revised manuscript and your response to the reviewers' comments. Your revised manuscript is also likely to be sent to reviewers for further evaluation.

Sincerely,

Inna Lavrik

Associate Editor

PLOS Computational Biology

Florian Markowetz

Deputy Editor

PLOS Computational Biology

Reviewer's Responses to Questions

**Comments to the Authors:**

Reviewer #1: In general, this is a very well written manuscript and the topic is certainly of interest to the readership of PLOS Comput. Biol. The computational method STAMP developed by the authors is envisioned to be a useful tool for the community for image analysis of apoptosis events, especially in the context of 3D culture. The authors not only clearly displayed the development of this algorithms, but also demonstrated its robustness and versatility via several case studies/applications.

Here I have several comments regarding to this manuscript, which are listed based on page number:

Page 4, under application to quantify chemotherapy-mediated cytotoxicity: the authors observed apoptosis rate fluctuated around 5% under control condition (Figure 2C), especially around 20-50 h, is that a normal apoptosis rate the author would expect?

Page 5, under application to quantify T-cell mediated cytotoxicity: There seems to have some discrepancy between figure 3C and 4. In Figure 3C, coculture of IGR-Pub and CTL gave apoptosis rate around 15% at 20-30 h, while in Figure 4, the rate for interval 20-28 h is already over 40% (24% for 20-24 h+19% for 24-28 h). Seems like the authors are using the exact same set of videos for quantification, can the authors provide some explanation about this discrepancy?

Page 5, under application to quantify T-cell mediated cytotoxicity: the authors indicated “after 2 days on chip the T cell viability was reduced (around 90 %)”. It would be better if the authors could provide more details in how to measure the viability for T-cells, either in this paragraph or in method.

Page 5, under cancer-associated fibroblasts promote chemo-resistance: the authors need to provide statistical analysis for the four conditions in Figure 5, especially the conditions +dox/-CAF and +dox/+CAF. The curves looked different, however the difference might not be significant due to the error (curves were plotted using SEM, not SD). The same applied for Figure 4A and 4B.

Page 5, under spatiotemporal analysis of cytotoxicity death: it would be clearer to define “nearby cells” here, eg. how close the dead event would be considered as a nearby event. Is the definition “one third of the estimated average cell radius” applied to all the quantification? Would cell density significantly affect the outcome? Also it would be interesting to see the result for MDA-MB-231 ±dox/+CAF in main text or supplementary info. Based on the result discussed in previous section, CAF impaired doxorubicin dependent apoptosis and therefore the Pdeath would be expected to be similar as natural death.

Page 6, under the discussion for STAMP method, paragraph 2: Navg(t, TLAG), “N” should be italic

Page 7, under the discussion for biological implications, paragraph 1: “in vitro” should be italic. “The CTL clone (P62) used is this work kills autologous cells (IGR-Pub)”, should be “used in this work”

Page 8, methods for cell cultures: for cell line authentication, the authors presumably refer to STR profiling instead of SRT? Maybe spell out this abbreviation.

Page 9, methods for live cell imaging: it would be better if the author could provide more details in setting up the time-lapse image acquisition, eg, wavelength, filter, exposure, etc. And seems like the way to position the AIM-Biotech devices is a key step in experimental setting. It would be helpful if the authors could provide some photos in supplementary info about this setting, which will enable easy application of this method by other audience.

Page 12, the authors are using Mann-Whitney-Wilcoxon nonparametric test through the entire manuscript, however, the sample size (n=3) is too small to achieve any statistical meaning. As it is stated on the GraphPad’s website (https://www.graphpad.com/guides/prism/8/statistics/how_the_mann-whitney_test_works.htm), “If you have small samples, the Mann-Whitney test has little power. In fact, if the total sample size is seven or less, the Mann-Whitney test will always give a P value greater than 0.05 no matter how much the groups differ.” Also, in many figures, for example Figure 2C, it seems clear that MDA-MB-231 control 0-10, MDA-MB-231 doxorubicin 0-10/30-40 are different, however, due to the small sample size, there is no significant difference achieved. And therefore, the statement on page 4 “we never found statistically significant differences between automated and manual counting, indicating that the algorithm is validated with respect to a standard human-controlled quantification method” is not appropriate. The authors need to increase the sample size to support their statement. I would envision there would be significant difference between manual counts and automatic counts, can the author provide more details about why that would occur, and is there any improvement can be made. In addition, can the authors explain why they chose Mann-Whitney-Wilcoxon test instead of t test? And for the replicates, are they technical replicates (different views) or biological replicates (different samples)? I would recommend using biological replicates to achieve solid result.

Page 19, figure 2B, please define blue arrow. Presumably the authors are indicating CTL cells?

Also, I don’t find the info related to deposit of this software. It will be helpful if the authors could provide a link/user manual to guide the audience getting access to this tool.

Reviewer #2: Manuscript PCOMPBIOL-D-20-01728 described new video analysis algorithms to reveal drug- or cell-induced cytotoxic effects (apoptosis) on human cancer cells in time and in space. The new method, abbreviated STAMP, has shown its applications to study the kinetics of cell death. The unsuspected finding of "death transmissibility" unveiled by this method is very interesting and warrants further mechanism studies.

Overall, this manuscript has been well prepared. I recommend its publication in PLOS Computational Biology with no revisions.

**Have all data underlying the figures and results presented in the manuscript been provided?**

Reviewer #1: **No: **The authors did not provide detailed data yet for their analysis. Also, it would be helpful if the authors could provide a link for software deposit and guide the audience getting access to this tool.

Reviewer #2: Yes

PLOS authors have the option to publish the peer review history of their article (what does this mean?). If published, this will include your full peer review and any attached files.

Reviewer #1: No

Reviewer #2: No
---

## [Decision Letter · Decision Letter 1]

17 Feb 2021

Dear Prof. Martinelli,

Thank you very much for submitting your manuscript "Apoptosis mapping in space and time of 3D tumor ecosystems reveals transmissibility of cytotoxic cancer death" for consideration at PLOS Computational Biology. As with all papers reviewed by the journal, your manuscript was reviewed by members of the editorial board and by several independent reviewers. The reviewers appreciated the attention to an important topic. Based on the reviews, we are likely to accept this manuscript for publication, providing that you modify the manuscript according to the review recommendations.

Sincerely,

Inna Lavrik

Associate Editor

PLOS Computational Biology

Florian Markowetz

Deputy Editor

PLOS Computational Biology

[LINK]

Reviewer's Responses to Questions

**Comments to the Authors:**

Reviewer #1: The authors carefully addressed most of my comments and the manuscript got significant improved. It should be ready for acceptance for publication, here I just have a few minor points that I would like to further confirm with the authors:

For T cell quantification, seems like the method is via manually counting. As this manuscript is all about developing an algorithm to simplify video analysis, is it possible to establish an automatic way to streamline the process? Or what is the difficulty in developing such method? Please discuss more in the manuscript.

In terms of the definition about “nearby cells”, I am still a little confused, as I thought the parameter “r” restricts the event of nearby cells (death wake). While as the authors wrote in the response, it is included all the cells within the detection window? Can the authors provide more explanation towards this question?

Also for the discrepancy between manual and automated counting, to me there is significant difference between this two groups at certain time point (in figure 2c). Please perform statistical test and specify accordingly on the figure. Looks like the discrepancy is more likely to occur at early time point, is there any reason why automated counting has lower readout at early time point? Why the algorithm failed to pick apoptotic cells which could be easily identified via manual counting? Please provide more discussion about this discrepancy.

Reviewer #2: I have no further comments. The manuscript has been well revised with a high quality to be accepted by PLOS Computational Biology.

**Have all data underlying the figures and results presented in the manuscript been provided?**

Reviewer #1: Yes

Reviewer #2: Yes

PLOS authors have the option to publish the peer review history of their article (what does this mean?). If published, this will include your full peer review and any attached files.

Reviewer #1: No

Reviewer #2: No
---

## [Editor Report · Decision Letter 2]

12 Mar 2021

Dear Prof. Martinelli,

We are pleased to inform you that your manuscript 'Apoptosis mapping in space and time of 3D tumor ecosystems reveals transmissibility of cytotoxic cancer death' has been provisionally accepted for publication in PLOS Computational Biology.

Best regards,

Inna Lavrik

Associate Editor

PLOS Computational Biology

Florian Markowetz

Deputy Editor

PLOS Computational Biology

---

## [Editor Report · Acceptance letter]

25 Mar 2021

PCOMPBIOL-D-20-01728R2 

Apoptosis mapping in space and time of 3D tumor ecosystems reveals transmissibility of cytotoxic cancer death

Dear Dr Martinelli,

I am pleased to inform you that your manuscript has been formally accepted for publication in PLOS Computational Biology. Your manuscript is now with our production department and you will be notified of the publication date in due course.

With kind regards,

Katalin Szabo
